# Survivor Guilt as a Mediator Between Post-Traumatic Stress Disorder and Pessimism Schema After Türkiye-Syria Earthquake

**DOI:** 10.3390/bs15091199

**Published:** 2025-09-03

**Authors:** Selma Çilem Kızılpınar, Barış Kılıç-Demir

**Affiliations:** Department of Psychiatry, Adana City Research and Training Hospital, Adana 01370, Türkiye; baris.kilicdemir@saglik.gov.tr

**Keywords:** survivor guilt, pessimism schema, depression, post-traumatic stress disorder

## Abstract

This research examines the relationship between socio-demographic characteristics of the survivors and their changing social situations after the earthquake and post-traumatic stress disorder (PTSD) and survivor’s guilt. It also examines the mediating role of survivor guilt between early maladaptive schemas and PTSD severity. The study involved 127 people exposed to the 6 February 2023, southern and central Türkiye and northern and western Syria earthquake. Participants’ sociodemographic characteristics, changing living conditions due to the earthquake, and feelings and thoughts of survivor guilt were evaluated with online data forms. Young Schema-Short Form, Post-Traumatic Stress Disorder Symptom Control Checklist, Beck Depression Inventory, and Beck Anxiety Inventory were employed. There was a notable connection between survivor guilt, the pessimism schema, and the PTSD severity. Additionally, changes in the participants’ living conditions especially occurring during posttraumatic periods were associated with survivor guilt and PTSD levels. The mediation analysis indicated that survivor guilt partially mediates the relationship between the pessimism schema and PTSD severity. Focusing on the social conditions of survivors, pessimism schema, and survivor guilt will be beneficial in preventive interventions and treatment approaches for PTSD.

## 1. Introduction

A multitude of factors have been noted that contribute to the trajectory of post-traumatic psychological responses. Risk factors for the development of post-traumatic stress disorder (PTSD) briefly encompass age, gender, IQ level, education status, ethnicity, prior trauma experience, trauma type, continued perception of threat, access to the needed resources, and social support after trauma ([12]; [43]). Researchers delve into the factors contributing to PTSD’s etiology in the past and present. Moreover, it is beneficial to underscore a significant effect in the development of PTSD—specifically, the examination of cognitive distortions and early maladaptive schemas (EMS) derived from Schema Therapy ([58]). According to Schema Therapy, EMS is attributed to childhood traumatization and frustration in meeting fundamental childhood needs, which are characterized as self-defeating emotional and cognitive patterns that originate in early childhood and persist throughout the lifespan ([58]). Early negative experiences can cause people to develop negative cognitive styles, and this ultimately causes differences in their interpretation and perception of new traumatic experiences ([19]). In this context, it is associated with the etiopathogenesis of most of the psychiatric disorders ([3]). It is beneficial to consider that it is a transdiagnostic concept that indexes a general vulnerability to psychopathology. Various studies exist investigating the relationship between PTSD and EMS. In the literature, it has been stated that the scores of all 18 schema dimensions were found to be higher in individuals with PTSD than in individuals who did not develop PTSD, and the most notable schema dimensions were stated as mistrust (insecurity), self-sacrifice, unrelenting standards, and punitiveness ([1]; [3]). A recent review highlighted that nine Early Maladaptive Schemas related to the Disconnection/Rejection and Impaired Autonomy/Performance domains are commonly observed among trauma-experienced populations ([29]). A study examining the relationship between childhood trauma and Complex PTSD reported a mediating role for EMSs. The study particularly focused on the total EMS score and the Disconnection and Impaired Autonomy schema domains ([51]). However, specific schema dimensions that distinguished post-traumatic stress disorder from the non-clinical group have not been identified. There is a necessity for research dedicated to investigating schema dimensions as potential associations with clinical disorders.

PTSD presents with a wide range of emotions: anxiety, anger, fear, guilt, and shame ([9]; [30]). The feeling of guilt, in particular, has consistently been connected to the emergence, persistence, and severity of symptoms related to PTSD. Guilt is found to be positively associated with symptoms such as re-experiencing traumatic events and avoidance symptoms, as well as with an overall measure of how severe the PTSD symptoms are ([9]; [21]). Innocent survivors may feel a varying degree of guilt, although they do not have any fault or offense after rescue. It is conceptualized as ‘survivor guilt’. It is positioned between external reality and inner consciousness and is describable as the coexistence of the survivor’s objective innocence and a subjective feeling of guilt ([28]; [32]; [38]). Even though it is generally associated with post-traumatic stress disorder, it can manifest independently of the existence of post-traumatic stress disorder. The theoretical basis of the concept of survivor guilt is initially based on the experiences of survivors of Holocaust trauma and Hiroshima ([16]; [33]; [34]). It was also subsequently reported among survivors of war, violent crimes, natural disasters, and some mortal illnesses ([17]; [37]). In the literature, survivors’ feelings of guilt were associated with increased severity of the psychopathology and resistance to treatment ([24]; [33]). Despite several proposed explanations, there remains a gap in our understanding of survivor guilt. A focused exploration of the etiopathogenesis of survivor guilt would be beneficial due to its complications. To our knowledge, no study has yet examined the relationship between early maladaptive schemas, survivor guilt, and PTSD. Addressing this gap in the literature may contribute to the identification of appropriate therapeutic targets within schema therapy and other treatment modalities that focus on cognition, emotion, and experiential processing. As a key research question in our study, we investigated the mediating role of survivor guilt in the relationship between Early Maladaptive Schemas (EMS) and PTSD. Survivor guilt may serve as a cognitive-affective bridge between underlying EMS and PTSD symptoms. Individuals with activated and elevated early maladaptive schemas may be susceptible to experiencing guilt after surviving a disaster, which may, consequently, intensify trauma-related distress and symptom severity.

On 6 February 2023, two significant earthquakes were experienced, a 7.8 magnitude earthquake and a 7.7 magnitude earthquake, in southern and central Türkiye and northern and western Syria, and over 570 aftershocks were recorded within 24 h ([49]). Big cities collapsed with lots of buildings, thousands of people died, and local people lost their homes and loved ones. After the earthquake, we observed many earthquake survivor interviews since our hospital received many referrals. During our psychiatric interviews with earthquake survivors, we observed individuals expressing thoughts such as ‘I should have been the one who did not survive, not him’ and ‘he did not deserve to die.’ These statements vividly demonstrated the profound impact of survivor guilt on those who experienced the earthquake. Therefore, we aimed to investigate factors associated with survivor guilt. The goals of the present study were (1) to clarify the mediation relationship between PTSD, EMS, and survivor guilt and (2) to examine the factors related to survivor guilt. Examining these questions may provide the development of appropriate therapeutic interventions.

## 2. Materials and Methods

The participants were recruited between 20 June and 10 September 2023, and participation was entirely voluntary, with no incentives offered. Participants were recruited via the website link, and they answered questions anonymously. Only people who were in the ten most affected cities of Türkiye on 6 February 2023 and survived after the earthquake participated in the study. The provinces we included within the extent of the study were determined by considering the provinces where the government declared a state of emergency after the earthquake ([42]). The data of those who had fully completed the study were analyzed. The participants who gave incomplete answers to more than ten items were excluded from the study. Of the 220 individuals who started the survey, 127 completed it (57.7%). Ethics committee approval was received by the hospital Ethics Committee (with the date 27 April 2023 and decision number 2534). The procedures used in the current study are designed for the Declaration of Helsinki. Informed consent was obtained from all participants.

### 2.1. Instruments

Sociodemographic Characteristics and Changing Living Conditions Data Form: Following the earthquake, significant alterations occurred in people’s living conditions, with some homes sustaining severe damage, and some people had to relocate their homes and stay with their relatives. Tent and container cities were constructed, and the survivors had to move there, besides becoming prolonged residences for some survivors. The survey encompassed inquiries about the patient’s sociodemographic information, such as age and education level, and post-earthquake experiences, evaluating aspects like the extent of their house damage, loss of their relatives/friends, relocation, residency situation in the first month after the earthquake, and current living arrangements. The assessment regarding the level of damage to their home included the damage levels determined by the government. Data forms were collected online via Survey Monkey ([52]). Detailed information on all items is provided in the statistical analysis section.

Survivor Guilt Measurement: The authors defined three statements to assess survivor guilt, encompassing feelings of guilt about surviving, thoughts of not deserving survival, and feelings of responsibility for negative earthquake consequences. Participants rated these feelings and thoughts over the last month on a scale from 1 to 5 (1: none, 5: extremely true). The items were constructed based on a comprehensive review of the literature on survivor guilt in trauma contexts ([8]; [13]; [39]; [43]). The items were designed to capture key cognitive–emotional elements of survivor guilt as conceptualized in the literature. A pilot application was conducted with ten individuals to test the understandability and comprehensibility of the items prepared by the authors. And no misunderstandings were reported. The internal consistency of the three-item scale, as measured by Cronbach’s alpha, was 0.86, indicating good reliability. The sum of the answers to three statements was calculated as the total survivor guilt score.

Beck Depression Inventory-II (BDI) was used to measure the severity of survivors’ depression. It has 21 items, and each item is scored on a four-point scale ranging from 0 (absent) to 3 (severe) ([4]). In the Turkish adaptation study, participants were asked to evaluate their depressive symptoms based on experiences during the past week. A total score of 17 or higher was accepted as the cut-off point for clinical depression ([22]). The BDI has demonstrated high internal consistency and is widely used in clinical populations due to its robust psychometric properties. As a self-report measure, it enables practical and time-efficient application.

Beck Anxiety Inventory (BAI) was used to measure the severity of survivors’ self-reported anxiety by administering it. It is a 4-point Likert-type questionnaire, with 21 items ([45]). In the Turkish adaptation study, participants were instructed to assess their symptoms based on the past week, and a cut-off score of 16 was established ([48]). The Turkish version of the BAI has been demonstrated to be a valid and reliable instrument across various populations, and it was considered both appropriate and easy to administer for assessing anxiety in the disaster-affected study sample.

Young Schema-Short Form version-3 (YSQ-3); was developed for the evaluation of the EMS and consists of 90 items and contains 18 schema dimensions ([58]). Participants rated each item on a 6-point Likert-type scale (1 = Completely false for me, 6 = Describes me perfectly) regarding how they have felt over the past year. Higher scores indicate higher levels of early maladaptive schema. The validity and reliability study of the short form of the scale was performed, and a 14-factor structure (14 schema dimensions) was determined. These dimensions include Emotional Deprivation, Failure, Pessimism, Social Isolation/Insecurity, Emotional İnhibition, Admiration Seeking, Enmeshment/Dependency, Insufficient Self-Control, Self-Sacrifice, Abandonment, Self Punitiveness, Defectiveness, Vulnerability, and Unrelenting Standards dimensions. In the literature, the Cronbach’s alpha internal consistency for the Young Schema-Short Form version-3 scale’s schema dimensions has been found to vary between 0.63 and 0.80 ([44]). For the current study, the Cronbach’s alpha value of the Young Schema-Short Form version-3 scale’s schema dimensions was observed to range between 0.65 and 0.85. 

PTSD Symptom Control Checklist (PCL-5) was developed for the self-report measurement of the PTSD symptom severity considering the DSM-5 meeting criteria ([55]). It is a 20-item and a Likert-type 5-point rating self-report questionnaire. Each item ranges from 0 (“not bothered at all”) to 4 (“bothered extremely”), with responses to be based on the last month’s experiences. The Turkish version of PCL-5 was reported to be valid and reliable and suggested a cut-off score of ≥47 for PTSD diagnosis, with 0.76 sensitivity and 0.69 specificity ([7]). Four-factor structure was determined; items 1 to 5 for the Intrusion Dimension, items 6 and 7 for the Avoidance Dimension, items 8 to 14 for the Negative Alterations in Cognitions and Mood Dimension, and items 15 to 20 for Alterations in Arousal and Reactivity dimension.

### 2.2. Statistical Analysis

The distribution of variables was checked with the Kolmogorov–Smirnov test. Parametric tests were used if the assumptions were provided, and non-parametric tests were used if they were not. In addition, the Pearson Chi-Square Test was used for comparisons between categorical variables. Finally, the Pearson or the Spearman Test carried out the correlation analyses. Mediation analysis was conducted via linear regression analysis to assess mediating relationships between early maladaptive schemas and survivor guilt and PTSD. All statistical analyses were performed with the SPSS version 23.0 package program, and the significance level was accepted as *p* < 0.05 ([31]). The assumptions for mediation analyses were tested using linear regression analyses. Hayes process macro for SPSS was used in the mediation analysis (model 4) ([20]).

## 3. Results

We asked 220 individuals in our research, and 127 participants agreed to complete the study. The average age of the group was 35.97 ± 9.04. Patients were evaluated at approximately day 188.92 ± 13.46 after the earthquake (min: 136–max: 219). See Table 1 for detailed review.

### 3.1. Correlation Analysis

Correlation analyses revealed a significant relationship between the PTSD score and various schema dimensions. Since clinical depression can affect schema-related measurements, correlation analysis was performed only in non-depressed patients with a BDI value below 17 (n = 80). A moderately significant relationship existed between the PCL-5 total score and pessimism schema (r = 0.463, *p* = 0.000) (Table 2). No correlation was found between other schemas and the PCL-5 score. There was no relationship between PCL-5 score and age, time duration after the earthquake, and total education years of the survivors (Table 3). According to the examination of the association between level of PTSD and survivor guilt, there was a relationship between the total survivor guilt score and the Intrusion Subscale of PCL-5 (r = 0.481, *p* = 0.000), the Negative Alteration Subscale (*p* = 0.000, r = 0.480), the Avoidance Subscale of PCL-5 (r = 0.587, *p* = 0.000), the Arousal Alterations Subscale (*p* = 0.000, r = 0.508), and PCL-5 total score (r = 0.561, *p* = 0.000). There was a moderately significant relationship found between the level of survival guilt and BDI total score (r = 0.513, *p* = 0.000) and BAI total score (r = 0.460, *p* = 0.000). A weak relationship was found between age and survival guilt (r = −0.195, *p* = 0.032) (Table 3). There is no relationship between survivor guilt and time duration after the earthquake and total education years.

Since clinical depression can affect schema-related measurements, correlation analysis was performed only in non-depressed patients with a BDI value below 17 (n = 82). There was a low level of relationship between survivor guilt and Pessimism Schema (r = 0.328, *p* = 0.003), Dependency Schema (r = 0.337, *p* = 0.002), Failure Schema (r = 0.298, *p* = 0.007), Social Isolation Schema (r = 0.233, *p* = 0.037), Abandonment Schema (r = 0.286, *p* = 0.010), and Defectiveness Schema (r = 0.244, *p* = 0.029). Notably, in individuals without depression, a discernible moderate association was observed between survivor guilt and pessimism and dependency schemas (Table 4).

### 3.2. Difference Analysis

In this part of the study, significant relationships were observed between survivor guilt and several variables reflecting the survivors’ current life conditions and clinical manifestations (Table 5). Significant differences were observed in several factors, according to the analysis. These included participants with moderately or severely damaged homes (U = 691.000, Z = −4.149, *p* = 0.000), participants who are not continuing to reside in their former homes in the last three months (U = 945.000, Z = −3.823, *p* = 0.000), individuals who experienced the loss of a relative or significant other (U = 1139.500, Z = −3.024, *p* = 0.002), participants whose homes were destroyed by the earthquake (U = 261.500, Z = −3.252, *p* = 0.001), similarly participants whose relatives’ or friends’ homes destroyed (U = 1355.000, Z = −3.279, *p* = 0.001), individuals who had to relocate to a different city (U = 1120.500, Z = −3.005, *p* = 0.003), participants diagnosed with depression according to the BDI (U = 1052.500, Z = −4.400, *p* = 0.000), and diagnosed with anxiety disorder according to BAI (U = 1187.000, Z = −4.302, *p* = 0.000). These identified variables collectively represent essential determinants influencing the manifestation of survivor guilt in the context of the survivors’ present or post-earthquake circumstances. No significant difference was detected between the groups regarding gender and place to stay in the first month after the earthquake.

### 3.3. Mediation Analysis

In this study, it was aimed to test whether survivor guilt mediates the relationship between pessimism and PTSD. Initial correlation and regression analysis indicated that all variables have a significant relationship. The analysis revealed that the relationship between variables met the essential assumptions for conducting mediation analysis ([50]). Results revealed a notable connection between survivor guilt, the Pessimism Schema, and the PCL-5 score. Specifically, the pessimism schema significantly affected the Survivor Guilt score (*p* = 0.000, β = 0.51, se = 0.03, t = 6.49, CI = 0.13–0.25), and the R square value was determined as 0.255. R^2^ values show the amount of variance explained by the model. According to the results, it can be said that the pessimism schema explains 25.5% of the variability of survivor guilt. Survivor guilt also had a significant effect on PTSD level (measured by PCL-5) (*p* = 0.000, β = 0.65, SE = 0.65, t = 7.74, CI = 3.73–6.30) with 32.8% of the variability in PTSD symptom severity (measured by PCL-5) attributed to survivor guilt (R^2^ = 0.328). Furthermore, the Pessimism schema also had an effect on PTSD level (measured by PCL-5) (*p* = 0.000, β = 0.59, SE = 0.25, t = 8.13, CI = 1.54–2.52), explaining 34.9% of the variability in PTSD symptom severity (R^2^ = 0.349). According to analysis, the pessimism schema had a significant positive direct effect on PTSD severity (measured by PCL-5) (*p* = 0.000, β = 0.41, SE = 0.27, t = 5.24, CI = 0.87–1.94) and survivor guilt significantly mediates this relationship (indirect effect, *p* = 0.000, β = 0.18, SE = 0.04, t = 1.41, CI = 0.10–0.26) (Table 6 and Figure 1). The mediation analysis indicated that survivor guilt partially mediates the relationship between the pessimism schema and PTSD severity, as evidenced by the reduction in the beta coefficient in the indirect model.

## 4. Discussion

Following the traumas, many survivors struggle with comprehending the trauma they have been exposed to, and sometimes guilt feelings may emerge after significant losses. This phenomenon is known as survivor guilt, involving survivors’ emotional distress and negative self-evaluation. Essentially, survivor guilt manifests when those who survive think of themselves as responsible for the harm experienced by others despite lacking control over the situation ([47]; [53]). Although there are various studies on this subject, the etiopathogenesis of survivor’s guilt is unclear and controversial ([35]). Our investigation aimed to examine survivor guilt as a mediator role between schema dimensions and PTSD severity and asses the possible factors related to survivor guilt. To our knowledge, this was the first study to assess EMS, survivor guilt, and PTSD severity following a natural disaster. Previous research has shown the association of EMS with the development, maintenance, and recurrence of severe mental disorders and that assessment of EMS is beneficial to identifying at-risk groups and preventing clinical disorders ([11]; [46]). Specifically, there are also studies exploring the connections between PTSD and EMS. However, in these studies, specific schema dimensions were not evaluated one by one; either a general score or schema domain evaluations were made, or a specific schema dimension could not be determined. In a study among Vietnam veterans, no specific schema dimension could be determined. More broadly, Emotional Inhibition, Self-Sacrifice, Entitlement/Grandiosity, Insufficient Self-Control, Failure, Mistrust/Abuse, Social Isolation/Alienation, Defectiveness/Shame, and Emotional Deprivation were significant characteristics of patients who have PTSD ([12]). Similarly, among prison officers, most of the schema dimensions were at a higher level ([6]). Among women with a history of interpersonal trauma, the Disconnection and the Impaired Autonomy schema domains were related to PTSD ([23]). In our study, the relationship between schema dimensions and PTSD level was investigated in patients without clinical depression, and a significant relationship was shown only between the pessimism schema and PTSD level. It was found that the contribution of the pessimism schema to the variance in the development of PTSD was 34.9%. This relationship is remarkable. In addition, it is valuable to demonstrate the relationship with a specific EMS in the study, one step further than previous studies. Based on our study findings, it can be suggested that individuals with pessimism schemas may be susceptible to developing PTSD or that the pessimism schema might be linked with the clinical manifestation of PTSD.

PTSD and survivor guilt are often described as closely intertwined phenomena. The emotion of guilt—particularly survivor guilt—has been associated with PTSD development and symptom severity in the literature ([2]; [5]; [21]; [40]). Despite numerous studies demonstrating an association between guilt and PTSD, the underlying mechanisms of this relationship remain unclear. The literature suggests that guilt and PTSD may co-occur as consequences of trauma, without a direct causal link, or that other related or overlapping psychological constructs may mediate their association ([26]). Another perspective on the relationship between PTSD and guilt suggests that survivor guilt may influence PTSD symptoms indirectly, through its impact on gratitude and perceived social support. In other words, individuals who feel survivor guilt may experience lower levels of gratitude or support, which in turn may increase the severity of PTSD symptoms ([53]). The relationship among schema dimensions, PTSD, and survivor guilt has not undergone comprehensive examination. In notable studies on this subject, a significant association was discerned between post-traumatic guilt and the emergence of PTSD, characterizing it as guilt-based PTSD. These studies underscored the potential influence of individual factors, specifically pre-traumatic schemas, on the manifestation of survivor guilt ([9]; [27]). Similarly, a recent study stated that negative self-evaluations, such as post-traumatic shame and self-blame, are factors that contribute to the severity of PTSD symptoms following traumatic events ([56]). Although it has been reported in the literature that post-traumatic self-blaming thoughts may mediate the relationship between early maladaptive schemas and PTSD, a specific relationship with EMS has not been stated. Our study findings align with prior research that has demonstrated the association between guilt and PTSD. This relationship was significantly important for all four subscales evaluated with PCL-5.

Furthermore, the pessimism schema was associated with PTSD level independently of the depression level, and there was a significant relationship between the pessimism schema and survivor’s guilt. Mediator variable analysis showed that 25.5% of the variability of survivor guilt is explained by the pessimism schema, and the pessimism schema explains 34.9% of the variability of PTSD symptom severity. It was also found that survivor’s guilt partially mediated the effects of the pessimism schema on PTSD symptom severity. According to the results, survivor guilt may be a vital pathway from pessimism schema to PTSD symptom severity due to an increased negative self-appraisal. Our study is important in that it showed the relationship between EMS, PTSD, and survivor guilt. Moreover, the co-occurrence of survivor’s guilt and the pessimism schema suggests heightened vulnerability to PTSD. Our findings indicate that survivor guilt and pessimism schema may be an important intervention point for clinicians aiming to reduce PTSD symptom severity. Such insights are particularly crucial for therapeutic approaches like cognitive-behavioral therapy and schema therapy, offering specific schemas for therapists to address during treatment.

While our primary focus centered on exploring the connections between survivor guilt, EMS, and PTSD, our study has also yielded other significant findings. The most prevalent psychiatric pathologies that manifest post-traumatic period are PTSD, anxiety disorder, and major depression ([10]; [57]). In our study, we also investigated the rate in the sample of PTSD, depression, and anxiety disorders in earthquake survivors. Through assessments considering scale cut-off values, according to our results, depression was detected in 35.4% (n = 45) of the participants, anxiety disorder was detected in 46.5% (n = 59), and PTSD was detected in 33.9% (n = 43). These psychopathologies generally indicate comorbid conditions with each other, not isolated pathologies. In our study, the rate of these disorders appears to be lower than the rates reported in the existing literature concerning post-traumatic psychopathologies ([41]). Obtaining different results regarding post-disaster psychopathology rates may be due to methodological differences, time after the disaster, and the nature of the sample ([54]). Another possible reason may be that an essential part of the participants left the study before filling out the scales completely despite initially agreeing to participate. Considering that patients with decreased attention, low motivation, loss of interest, and perhaps more severe symptoms may have left the study early, we might have failed to detect people with symptoms at a diagnostic level. Consequently, the rate of mental disorders in our study may be underestimated. We think that comprehensive studies are needed to determine the mental disorder rate after the earthquake.

In our study, younger participants reported significantly higher levels of intrusion and avoidance symptoms related to the traumatic event. The negative correlation we identified between age and both avoidance and intrusion symptoms is consistent with prior literature indicating greater PTSD severity among younger individuals ([14]). Authors attributed this pattern to younger individuals having more limited life experience and coping resources. These findings highlight the importance of age-sensitive interventions following the trauma.

A pivotal aspect addressed in this study is the exploration of factors that could potentially be associated with survivor guilt. Pre-traumatic, peri-traumatic, and post-traumatic factors associated with PTSD have been identified in the literature ([43]). However, there are limited studies in the literature investigating clinical and sociodemographic factors related to survivor guilt, especially after natural disasters. Mood disorders (i.e., depression), decreased self-esteem, limited social support, individual characteristics, the type of event(s), developmental processes, the significance of the trauma, and sociocultural factors were associated with the destructive impact of a traumatic experience and survivor guilt ([39]). In our study, the relationship between survivor guilt and several variables reflecting the survivors’ current life conditions and clinical manifestations is as follows. 1. experiencing moderate or severe damage to the house they lived in before the earthquake; 2. having one’s house collapsed after the earthquake; 3. having at least one of one’s relatives or special one’s houses destroyed after the earthquake; 4. undergoing relocation after the earthquake; 5. experiencing at least one death of relatives or special others after the earthquake; 6. not having a chance to return to pre-earthquake home after the acute phase of the earthquake has passed, that is, in the last three months; 7. existence of depression or anxiety disorder. The result that the experience of loss is associated with survivors’ guilt is consistent with publications that emphasize the relationship between grief and survivors’ guilt ([15]). We underscore the importance of grief interventions in the management of survivor guilt. According to our results, no significant relationship was found between survivor guilt, the time duration after the earthquake, the existence of psychiatric disorder history before the earthquake, and the place to stay in the first month after the earthquake. However, a weak but statistically significant negative correlation was found between age and survivor guilt, suggesting that younger individuals experienced higher levels of guilt. This finding is consistent with negative correlations between age and intrusion and avoidance and align with previous research showing that younger individuals often exhibit more intense trauma responses ([14]). These age-related differences underscore the necessity of age-sensitive interventions in the aftermath of large-scale trauma. An important point we seek to emphasize is that post-traumatic changes in living conditions, rather than pre-existing factors like living conditions before the earthquake and conditions during the initial phase of the earthquake, are significantly associated with the severity of survivor guilt and PTSD symptoms. There was no relation between the time duration after the earthquake and PTSD symptom level and survivor guilt. This finding may be related to the timing of our assessment, which was conducted approximately nine months after the earthquake. Trauma responses should be examined at both earlier and later stages following the traumatic event to better understand the relationship between time duration and survivor guilt and PTSD. The results we obtained from our study regarding survivor guilt revealed the importance of changing life conditions and events, especially post-traumatic factors. The post-earthquake alterations in individuals’ living conditions emerge as crucial determinants for the emergence of psychopathologies. The responsibility for addressing this lies with the government. Timely engagement with the survivor after the earthquake and the regulation of social conditions will be beneficial in reducing disability due to psychopathologies.

### Limitations

A key limitation of the study concerns the sample size and its representativeness. Although participants were recruited from the ten provinces most severely affected by the earthquakes, the study population was not proportional to population size or the distribution of disaster impact across these regions. Additionally, it is essential to underscore that our investigation did not extend to an examination of the broader trauma histories of the participants and, notably, lacked a comparative analysis with a control group unexposed to the earthquake. The relatively modest sample size and insufficient qualified population distribution limit the findings’ generalizability. Furthermore, the study focused entirely on individuals exposed to the disaster and did not include a control group or an assessment of participants’ history of another trauma. Future research should aim to use larger, more representative samples and include comparison groups, as addressing these methodological gaps may yield more substantial and generalizable insights. Furthermore, due to the cross-sectional design and the current sample characteristics (like sample size), robustness or heterogeneity analyses (e.g., by age, gender, etc.) were not performed. In future studies conducted with larger samples and longitudinal designs, the use of such analyses would be beneficial for exploring subgroup differences and confirming the stability of the observed relationships.

It is a limitation that a valid and reliable scale regarding survivor guilt was not used since no scale in Turkish expressed survivor guilt after natural trauma. In studies, the interpersonal guilt scale was mostly used to employ for assessing survivors’ guilt ([53]), but there is no Turkish adaptation study of this scale ([18]; [36]). Another scale used in studies to assess survivor guilt is the Bereavement Guilt Scale ([40]). A validity and reliability study of the Turkish version of the Bereavement Guilt Scale was performed ([25]). However, the items of the scales inadequately capture the guilt experienced with fatalities ensuing from natural disasters, such as earthquakes. However, we accept that the lack of use of a valid and reliable scale is an important limitation. There is a need to develop scales to evaluate people’s feelings of guilt after natural disasters such as earthquakes using valid methods.

In conclusion, elucidating the interrelation between EMS, survivor guilt, and PTSD has significant potential for enhancing the therapeutic interventions designed for individuals with PTSD. Beyond its theoretical implications, a discerning comprehension of the personal and environmental factors contributing to both PTSD and survivor guilt enables the formulation of more nuanced and tailored treatment modalities. Our findings suggest the usefulness of cognitive behavioral interventions in the context of the pessimism schema to modify existing core beliefs and decrease subsequent symptomatology in adult survivors of interpersonal trauma. The outcomes of this study underscore the predictive efficacy of survivor guilt and the Pessimism schema in the context of PTSD. Moreover, empirical evidence of the study indicates that the losses, both in terms of property and beloved individuals, during the acute phase and along with the subsequent amelioration in living conditions post the acute process, play an essential role compared to sociodemographic factors in survivor guilt and PTSD.

## Figures and Tables

**Figure 1 behavsci-15-01199-f001:**
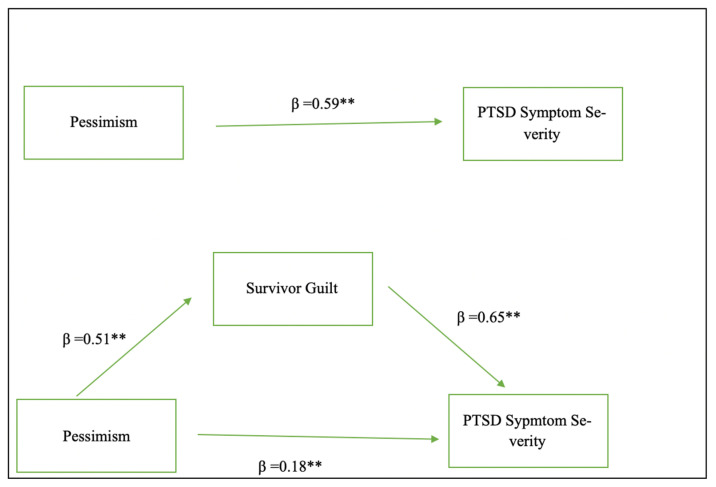
Mediation model of the relationship between pessimism, survivor guilt, and PTSD symptom severity. ** *p* = 0.00.

**Table 1 behavsci-15-01199-t001:** Participants’ sociodemographic characteristics and clinical features.

Feature	Patient Group (n)	%
Marital status Married	88	69
Gender Female	89	70
Occupational Status: Employed	106	83
The presence of comorbid medical disorder, Yes	29	23
Psychiatric disorder history before the earthquake, Yes	34	22
Admission to psychiatric outpatient clinic after the earthquake Yes	22	17
Damage level of the survivors’ house		
No damage	58	37
Low damaged	42	27
Moderately damaged	9	6
Highly damaged or destroyed	17	11
Place to stay in the first month after the earthquake		
Former home	24	19
With relatives/friends	66	52
Relocated to another house	14	11
Tent/container	7	6
Facilities provided by the government/other support organization	14	11
Hospital	2	2
Survivors who have a relative or special others who lost their home, Yes	70	56
Survivors who had to relocate the city they lived before	37	29
Place of stay for the last 3 months		
Former home	91	72
Relocate and living to another house	18	14
With relatives/another house (no private house)	14	11
Container/tent cities	4	3
Survivors who have a death relative or special others due to earthquake Yes	38	30
Total score of more than 17 on BDI	45	35
Total score of more than 16 on BAI	59	47
Total score of more than 48 on PCL-5	43	34
Continuous Variables		
Age (years)	35.97 ± 9.04	
Time duration between earthquake and the assessment procedure	188.92 ± 13.46	
Education years	16.24 ± 3.03	
Survivor Guilt Score	4.20 ± 2.24	
Beck Anxiety Scale (BAI)	17.02 ± 13.49	
PTSD Symptom Control Checklist (PCL-5)	38.48 ± 20.41	
PCL-5 Intrusion subdimensions score	10.38 ± 5.41	
Avoidance subdimensions score	3.85 ± 2.19	
Negative Alterations in Cognition and Mood Intrusion subdimensions score	13.24 ± 7.43	
Arousal and Reactivity subdimensions score	13.00 ± 7.43	

Categorical variables presented as n and %, continuous variables as Mean ± SD.

**Table 2 behavsci-15-01199-t002:** Spearman correlation analysis between PTSD scores and early maladaptive schemas in non-depressed participants (n = 80) ^a^.

	1	2	3	4	5	6	7	8	9	10	11	12	13	14	15
1. PCL5	1.000														
2. Emotional Deprivation sch.	−0.046	1.000													
3. Failure schema	0.008	0.355 **	1.000												
4. Pessimism	0.463 **	0.408 **	0.366 **	1.000											
5. Social Isolation/Insecurity	0.174	0.519 **	0.441 **	0.547 **	1.000										
6. Emotional İnhibition	0.035	0.397 **	0.459 **	0.453 **	0.464 **	1.000									
7. Admiration Seeking	0.104	0.303 **	0.257 *	0.399 **	0.597 **	0.272 *	1.000								
8. Enmeshment/Dependency	0.195	0.400 **	0.705 **	0.407 **	0.454 **	0.508 **	0.290 **	1.000							
9. Insufficient Self-Control	0.186	0.265 *	0.176	0.339 **	0.534 **	0.315 **	0.438 **	0.167	1.000						
10. Self-Sacrifice	0.150	0.249 *	0.199	0.265 *	0.237 *	0.263 *	0.437 **	0.325 **	0.242 *	1.000					
11. Abandonment	0.142	0.531 **	0.557 **	0.440 **	0.585 **	0.421 **	0.498 **	0.536 **	0.350 **	0.454 **	1.000				
12. Self Punitiveness	0.124	0.267 *	0.164	0.469 **	0.516 **	0.433 **	0.540 **	0.243 *	0.498 **	0.551 **	0.379 **	1.000			
13. Defectiveness	0.060	0.532 **	0.661 **	0.525 **	0.571 **	0.535 **	0.235 *	0.587 **	0.176	0.100	0.533 **	0.195	1.000		
14. Unrelenting Standards	0.071	0.242 *	0.132	0.455 **	0.391 **	0.255 *	0.400 **	0.161	0.349 **	0.343 **	0.192	0.437 **	0.228 *	1.000	
15. Vulnerability	0.169	0.572 **	0.305 **	0.561 **	0.645 **	0.482 **	0.574 **	0.463 **	0.361 **	0.408 **	0.547 **	0.603 **	0.461 **	0.310 **	1.000

* *p* < 0.05, ** *p* < 0.01. ^a^: correlation analysis was performed only in non-depressed patients (according to Beck Depression Inventory).

**Table 3 behavsci-15-01199-t003:** Spearman correlation analysis between PCL-5 and clinical and sociodemographic variables in all patients.

	1	2	3	4	5	6	7	8	9	10
1. Survivor Guilt										
2. PCL5 Intrusion Subscale	0.481 **									
3. PCL5 Negative Alterations	0.480 **	0.898 **								
4. PCL5 Avoidance	0.587 **	0.796 **	0.805 **							
5. PCL5 Arousal Alterations	0.508 **	0.793 **	0.823 **	0.872 **						
6. PCL5-Total Score	0.561 **	0.914 **	0.910 **	0.950 **	0.943 **					
7. BAI-Total Score	0.460 **	0.584 **	0.523 **	0.602 **	0.582 **	0.599 **				
8. BDI-Total Score	0.513 **	0.541 **	0.561 **	0.466 **	0.552 **	0.573 **	0.669 **			
9. Age	−0.195 *	−0.243 **	−0.076	−0.270 **	−0.094	−0.153	−0.098	−0.101		
10. Time duration after earthquake	−0.129	0.047	−0.010	0.103	0.029	00.30	0.078	0.002	0.039	
11. Education years	0.022	−0.141	−0.027	−0.151	−0.121	−0.105	−0.160	−0.075	0.065	−0.024

* *p* < 0.05, ** *p* < 0.01. PCL 5; PTSD Symptom Control Checklist, BAI; Beck Anxiety Inventory, BDI; Beck Depression Inventory.

**Table 4 behavsci-15-01199-t004:** Correlation between schema dimensions and survivor guilt in non-depressed patients (n = 82).

	1	2	3	4	5	6	7	8	9	10	11	12	13	14	15
1. Survivor guilt	1.000														
2. Emotional Deprivation	0.044	1.000													
3. Failure	0.298 *	0.355 **	1.000												
4. Pessimism	0.328 **	0.408 **	0.366 **	1.000											
5. Social Isolation/Insecurity	0.233 *	0.519 **	0.441 **	0.547 **	1.000										
6. Emotional İnhibition	0.213	0.397 **	0.459 **	0.453 **	0.464 **	1.000									
7. Admiration Seeking	0.221	0.303 **	0.257 *	0.399 **	0.597 **	0.272 *	1.000								
8. Enmeshment/Dependency	0.337 **	0.400 **	0.705 **	0.407 **	0.454 **	0.508 **	0.290 **	1.000							
9. Insufficient Self-Control	0.060	0.265 *	0.176	0.339 **	0.534 **	0.315 **	0.438 **	0.167	1.000						
10. Self-Sacrifice	0.118	0.249 *	0.199	0.265 *	0.237 *	0.263 *	0.437 **	0.325 **	0.242 *	1.000					
11. Abandonment	0.286 *	0.531 **	0.557 **	0.440 **	0.585 **	0.421 **	0.498 **	0.536 **	0.350 **	0.454 **	1.000				
12. Self-Punitiveness	0.064	0.267 *	0.164	0.469 **	0.516 **	0.433 **	0.540 **	0.243 *	0.498 **	0.551 **	0.379 **	1.000			
13. Defectiveness	0.244 *	0.532 **	0.661 **	0.525 **	0.571 **	0.535 **	0.235 *	0.587 **	0.176	0.100	0.533 **	0.195	1.000		
14.Unrelenting standards	0.033	0.242 *	0.132	0.455 **	0.391 **	0.255 *	0.400 **	0.161	0.349 **	0.343 **	0.192	0.437 **	0.228 *	1.000	
15. Vulnerability	0.103	0.572 **	0.305 **	0.561 **	0.645 **	0.482 **	0.574 **	0.463 **	0.361 **	0.408 **	0.547 **	0.603 **	0.461 **	0.310 **	1.000

* *p* < 0.05, ** *p* < 0.01.

**Table 5 behavsci-15-01199-t005:** Difference analysis related to survivor guilt.

		Survivor Guilt Score		
		N	MR	Sum of Ranks
Damage level of home				
Participants group whose house had low or no damage,		99	56.98	5641.00
Participant group whose house had moderately or more severe damage		26	85.92	2234.00
	Statistics	U = 691.000, Z = −4.149, *p* = 0.000		
Place of stay for the last 3 months				
The participants who are continuing to stay at former home before earthquake		91	56.38	5131.00
Other places		34	80.71	2744.00
	Statistics	U = 945.000, Z = −3.823, *p* = 0.000		
Survivors who have at least one death of a relative or special others due to earthquake				
Who have a death of a relative or special others		37	76.20	2819.50
Who have no death of a relative or special others due to earthquake		88	57.45	5055.50
	Statistics	U = 1139.500, Z = −3.024, *p* = 0.002		
Survivors whose home was destroyed due to earthquake				
Participants whose home was destroyed due to earthquake		10	93.35	933.50
Other participants		114	59.79	6816.50
	Statistics	U = 261.500, Z = −3.252, *p* = 0.001		
Survivors whose relatives/friends’ home was destroyed in the earthquake				
Participants whose relatives/friends’ home was destroyed due to earthquake		69	71.36	4924.00
Other participants		56	52.70	2951.00
	Statistics	U = 1355.000, Z = −3.279, *p* = 0.001		
Survivors who had to relocate from the city they lived before				
Participants who had to relocate from their city		36	76.38	2749.50
Others		89	57.59	5125.50
	Statistic	U = 1120.500, Z = −3.005, *p* = 0.003		
Place to stay in the first month after the earthquake				
The survivors who stayed to continue at former home and who stayed in their relatives’ or friends’ home		89	59.49	5294.50
The survivors who had to relocate to another house and who stayed in facilities provided by the government/other support organizations, containers, tents, or hospitals		34	68.57	2331.50
	Statistics	U = 1289.500, Z = −1.458, *p* = 0.145		
Existence of anxiety disorder				
BAI score was lower than cut off level (no anxiety disorder)		66	51.48	3398.00
BAI score was higher than cut off level (existence of anxiety disorder)		59	75.88	4477.00
	Statistics	U = 1187.000, Z = −4.302, *p* = 0.000		
Existence of Depression				
BDI score was lower than cut off level (no depression)		80	53.66	4292.50
BDI score was higher than cut off level (existence of depression)		45	79.61	3582.50
	Statistics	U = 1052.500, Z = −4.400, *p* = 0.000		
Gender				
Female		88	64.76	5699.00
Male		37	58.81	2176.00
	Statistics	U = 1473.000, Z = −0.959, *p* = 0.337		
Psychiatric disorder history before the earthquake				
Yes		34	64.26	2185.00
No		91	62.53	5690.00
	Statistics	U = 1504.00, Z = −0.273, *p* = 0.785		

N; Number of Participants, MR: Mean Rank, BAI; Beck Anxiety Inventory, BDI; Beck Depression Inventory.

**Table 6 behavsci-15-01199-t006:** Results of mediation analyses.

Effect	Pathway	Standardized Beta	Standard Error	Confidence Interval (95%)	F Value, t Value, *p* Value
Total	Pessimism → PTSD	0.59	0.25	1.53–2.52	F(1,123) = 66.01, t = 8.12, *p* = 0.00
İndirect	Pessimism → Survivor guilt → PTSD	0.18	0.04	0.10–0.26	F(2,122) = 49.06 *p* = 0.00
Direct	Pessimism → PTSD	0.41	0.27	0.87–1.94	t = 5.24, *p* = 0.00

## Data Availability

The data that support the findings of this study are available from the corresponding author upon your request.

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
