# Peer review of "Survivor Guilt as a Mediator Between Post-Traumatic Stress Disorder and Pessimism Schema After Türkiye-Syria Earthquake"

_behavsci, 2025, doi:10.3390/bs15091199_

Round 1
Reviewer 1 Report
Comments and Suggestions for Authors
The manuscript can be accepted in its present form for publication. However, it could be useful to reduce the number of correlations presented (I would avoid to put both the correlation with the total score of a scale and with its subscales) and I would also present the data about the depressed patients.
Author Response
Reviewer 1
The manuscript can be accepted in its present form for publication. However, it could be useful to reduce the number of correlations presented (I would avoid to put both the correlation with the total score of a scale and with its subscales) and I would also present the data about the depressed patients.
Thank you very much for your feedback. We accept that such an analysis about the depressive subgroup could be informative. However, the main objective of this study was to explore specific mechanisms among participants not affected by depression, in order to isolate psychological structures that are not confounded by clinical depression. Including data from the depressed group could digress from the primary focus of the study and introduce additional complexity beyond the initial objectives. Therefore, we did not add anything to the depressive group.
Reviewer 2 Report
Comments and Suggestions for Authors
The authors made an effort to collect a small sample of people who suffered and lived in areas affected by a strong earthquake in 2023. Months later, they used a non-probabilistic small sample collected online to study three variables and several sociodemographic data.
The authors tried to identify relationships between guilt, depression, anxiety, and trauma, to generate an explanatory model at the end.
As a general basis, the authors should consider adjusting their study, for example, since they report "prevalences" of PTSD, depression, and anxiety disorders, but such concepts do not apply to this study, since the sample and the size of the sample were not randomly collected.
 In any case, the arguments, speculations, and proposals they are stating apply only to the studied sample; they cannot be generalized to populations exposed to an earthquake, even to all people who have suffered human or material losses as a consequence of the disaster.
The use and proposal of "a mathematical model", as a consequence, is not feasible. This is due to the lack of sample power and other requirements necessary to justify and perform a model analysis.
I have several worries to propose:
- In the Instruments section, part 2.1 (page 3, lines 100 to 102), they mention three statements to measure Survivor Guilt. Where did the statements or items come from? How were they evaluated and validated?.
- On page 4, lines 110 to 117, please explain how you determined that 17 points would be the cutoff point for depression, and in which population. The same goes for the 16-point anxiety instrument. b) Also, please consider that these instruments are intended to measure symptoms in the last two weeks.
- On page 4, lines 125-126, the authors report a Cronbach´s alpha of 0.94 for the Young Schema Questionnaire-3 (YSQ-3). However, remember that to calculate the consistency, reliability, and validity of this scale, at least 450 participants would be needed, 5 for each item.
- On page 5, (Results section), Table 1: a) Please, delete the % sign and the median, mean, or SD legends from the lines. Express these details in a notation and the heading of the Table. b) Please round up or down to the nearest whole number in all the percentages.
First, could you move the quantitative variables to the bottom of the table? c) Please, delete "The survivors who..." at the Place to stay in the first month variable. d) It would be desirable to view the values for each factor of the three inventories applied. (here, or in a
separated table. - I suggest generating three different tables, one table for depression, one for anxiety, and another one for PTSD symptoms, comparing the outcome groups according to
to the respective cut-off point declared. - From lines 157 to 227: a) Please, delete any report or mention of variables without significance in the text of the article. b) The authors must state (and discuss) the type of relationship they observed between age and survival guilt. c) Also, they should discuss the relationships they showed between age and PCL5 Intrusion Subscale, and the PCL5 Avoidance subscale (Table 3).
- On page 7, lines 193 to 194, please explain and justify how the authors decided "moderate" correlations in Table 4. Otherwise, delete those comments. All the reported figures can be labeled as "low," but they are significant.
- On page 8, Table 5: a) Please state the N for this table. Also, unveil the meaning of
acronyms. b) Fix the word "Participiant". - On page 10, Table 7. a) This is a Figure, not a Table. b) The Beta value between Pessimism and PTSD Symptom Severity is missing? Or why is this part shown at the top of the figure?
- On page 11, lines 333-337, the authors mention specific percentages. Please explain and justify where the authors showed the evidence for these percentages. (As I said before, the authors report "prevalences" of PTSD, depression, and anxiety disorders, but such concepts do not apply for this study, since the sample and the size of the sample were not randomly collected.
- On page 12, the authors failed to explain and discuss the negative relationship they showed between age and other factors studied.
- On page 13, the Limitations section. The authors should show official figures of disaster in the ten most affected cities of Turkey, to give us an idea of how representative the collected sample is.
Author Response
Response to the Reviewer 2
We would like to thank you for your valuable evaluation and comments. We have made the necessary changes emphasized by the reviewers. Reviewer comments (in italics), our response to reviewers, and the changes made in light of reviewer comments (in bold) are provided below.
The authors made an effort to collect a small sample of people who suffered and lived in areas affected by a strong earthquake in 2023. Months later, they used a non-probabilistic small sample collected online to study three variables and several sociodemographic data.
The authors tried to identify relationships between guilt, depression, anxiety, and trauma, to generate an explanatory model at the end.
As a general basis, the authors should consider adjusting their study, for example, since they report "prevalences" of PTSD, depression, and anxiety disorders, but such concepts do not apply to this study, since the sample and the size of the sample were not randomly collected.
 In any case, the arguments, speculations, and proposals they are stating apply only to the studied sample; they cannot be generalized to populations exposed to an earthquake, even to all people who have suffered human or material losses as a consequence of the disaster.
The use and proposal of "a mathematical model", as a consequence, is not feasible. This is due to the lack of sample power and other requirements necessary to justify and perform a model analysis.
I have several worries to propose:
- In the Instruments section, part 2.1 (page 3, lines 100 to 102), they mention three statements to measure Survivor Guilt. Where did the statements or items come from? How were they evaluated and validated?.
There is currently no Turkish scale available that specifically measures survivors' guilt following natural traumas. Existing instruments used in previous studies do not adequately capture the guilt experienced after deaths caused by natural disasters such as earthquakes. Therefore, based on our review of the relevant literature and supported by our clinical observations, we developed three original items. A pilot study was conducted to assess the comprehensibility of these three items. Further details regarding these items are provided in the Instruments and Limitations sections of the manuscript.
We added to the instruments section the following part.
Survivor Guilt Measurement: The authors defined three statements to assess survivor guilt, encompassing feelings of guilt about surviving, thoughts of not deserving survival, and feelings of responsibility for negative earthquake consequences. Participants rated these feelings and thoughts over the last month on a scale from 1 to 5 (1: none, 5: extremely true). The items were constructed based on a comprehensive review of the literatüre on survivor guilt in trauma contexts [1,22-24]. The items were designed to capture key cognitive-emotional elements of survivor guilt as conceptualized in the literatüre. A pilot application was conducted with ten individuals to test the understandability and comprehensibility of the items prepared by the authors. And no misunderstanding were reported. The internal consistency of the three-item scale, as measured by Cronbach’s alpha, was 0.86, indicating good reliability. The sum of the answers to three statements was calculated as the total survivor guilt score.
We made changes in the Limitations section
It is a limitation that a valid and reliable scale regarding survivor guilt was not used since no scale in Turkish expressed survivor guilt after natural trauma. In studies, the interpersonal guilt scale was mostly used to employ for assessing survivors' guilt [38], but there is no Turkish adaptation study of this scale [56,57]. Another scale used in studies to assess survivor guilt is the Bereavement Guilt Scale [44]. A validity and reliability study of the Turkish version of the Bereavement Guilt Scale was performed [58]. However, the items of the scales inadequately capture the guilt experienced with fatalities ensuing from natural disasters, such as earthquakes. However, we accept that the lack of use of a valid and reliable scale is an important limitation. There is a need to develop scales to evaluate people's feelings of guilt after natural disasters such as earthquakes using valid methods.
- On page 4, lines 110 to 117, please explain how you determined that 17 points would be the cutoff point for depression, and in which population. The same goes for the 16-point anxiety instrument. b) Also, please consider that these instruments are intended to measure symptoms in the last two weeks.
These cut-off points are based on Turkish validity and reliability studies. These references were already mentioned, but the explanation may have been poor. The reference and sample type will be detailed.
We added to the Instruments section the following part.
Beck Depression Inventory-II (BDI) was used to measure the severity of survivors' depression. It has 21 items, and each item is scored on a four-point scale ranging from 0 (absent) to 3 (severe) [27]. In the Turkish adaptation study, participants were asked to evaluate their depressive symptoms based on experiences during the past week. A total score of 17 or higher was accepted as the cut-off point for clinical depression [28]. The BDI has demonstrated high internal consistency and is widely used in clinical populations due to its robust psychometric properties. As a self-report measure, it enables practical and time-efficient application.
Beck Anxiety Inventory (BAI); was used to measure the severity of survivors' self-reported anxiety by administering it. It is a 4-point Likert-type questionnaire, with 21 items [29]. In the Turkish adaptation study, participants were instructed to assess their symptoms based on the past week, and a cut-off score of 16 was established [30]. The Turkish version of the BAI has been demonstrated to be a valid and reliable instrument across various populations, and it was considered both appropriate and easy to administer for assessing anxiety in the disaster-affected study sample.
- On page 4, lines 125-126, the authors report a Cronbach´s alpha of 0.94 for the Young Schema Questionnaire-3 (YSQ-3). However, remember that to calculate the consistency, reliability, and validity of this scale, at least 450 participants would be needed, 5 for each item.
The validity and reliability study for the Young Schema Questionnaire-3 has previously been conducted in Turkish. The values indicated in the reliability study were stated. In our study, Cronbach's alpha internal consistency coefficient was calculated for 14 subdimensions. The relevant statement has been amended.
- On page 5, (Results section), Table 1: a) Please, delete the % sign and the median, mean, or SD legends from the lines. Express these details in a notation and the heading of the Table. b) Please round up or down to the nearest whole number in all the percentages.
First, could you move the quantitative variables to the bottom of the table? c) Please, delete "The survivors who..." at the Place to stay in the first month variable. d) It would be desirable to view the values for each factor of the three inventories applied. (here, or in a
separated table.
Table 1 has been revised. All % signs and statistical descriptors (mean, SD) were removed from the body of the table and clarified in the table note and column headings. Percentages were rounded to the nearest whole number. Quantitative variables were moved to the bottom of the table. ''The survivors who...” statement was changed. Subdimensions scores for the PCL-5 added in the bottom of table. Since the subscales of the BDI and BAI scales were not used in our analysis, these values were not added to avoid the complexity of the table
- I suggest generating three different tables, one table for depression, one for anxiety, and another one for PTSD symptoms, comparing the outcome groups according to
to the respective cut-off point declared.
We agree that comparing outcome groups based on clinical cut-off points can provide valuable insights. However, in our study design, the central aim was to examine the associations between early maladaptive schemas (EMS), PTSD, and survivor guilt while minimizing the potential confounding effect of depression. Therefore, rather than analyzing diagnostic subgroups separately, we focused specifically on a subsample of 88 participants who did not meet the clinical threshold for depression (BDI < 17). This approach allowed us to analyse the relationship between schemas and PTSD symptom level with excluding depression. Accordingly, we did not include separate tables comparing participants by diagnostic categories.
- From lines 157 to 227: a) Please, delete any report or mention of variables without significance in the text of the article. b) The authors must state (and discuss) the type of relationship they observed between age and survival guilt. c) Also, they should discuss the relationships they showed between age and PCL5 Intrusion Subscale, and the PCL5 Avoidance subscale (Table 3).
We reviewed throughout the text for non-significant relationships. The we discussed these relationships between age and PCL5 subdimensions. We added following paragraph to the discussion section
In our study, younger participants reported significantly higher levels of intrusion and avoidance symptoms related to the traumatic event. The negative correlation we identified between age and both avoidance and intrusion symptoms is consistent with prior literature indicating greater PTSD severity among younger individuals [54]. Authors attributed this pattern to younger individuals having more limited life experience and coping resources. These findings highlight the importance of age-sensitive interventions following the trauma.
- On page 7, lines 193 to 194, please explain and justify how the authors decided "moderate" correlations in Table 4. Otherwise, delete those comments. All the reported figures can be labeled as "low," but they are significant.
Checked and corrected
- On page 8, Table 5: a) Please state the N for this table. Also, unveil the meaning of
b) Fix the word "Participiant".
Checked and corrected
- On page 10, Table 7. a) This is a Figure, not a Table. b) The Beta value between Pessimism and PTSD Symptom Severity is missing? Or why is this part shown at the top of the figure?
In the revised manuscript, we corrected the label and now relabeled Table 7 as Figure 1. We also clarified the direct effect path from Pessimism to PTSD Symptom Severity within the figure by adding the missing standardized beta value.
- On page 11, lines 333-337, the authors mention specific percentages. Please explain and justify where the authors showed the evidence for these percentages. (As I said before, the authors report "prevalences" of PTSD, depression, and anxiety disorders, but such concepts do not apply for this study, since the sample and the size of the sample were not randomly collected.
We revised this statement and prefer terms like "rate" or "frequency in sample" to
"prevalence."
- On page 12, the authors failed to explain and discuss the negative relationship they showed between age and other factors studied.
Rather than focusing on the weak negative correlation found between guilt, PTSD and age in our study, we wished to highlight the post-traumatic factors with which we identified stronger associations. In line with your suggestion, we have expanded the discussion section of the manuscript. The changed text is as follows
In our study, younger participants reported significantly higher levels of intrusion and avoidance symptoms related to the traumatic event. The negative correlation we identified between age and both avoidance and intrusion symptoms is consistent with prior literature indicating greater PTSD severity among younger individuals [54]. Authors attributed this pattern to younger individuals having more limited life experience and coping resources. These findings highlight the importance of age-sensitive interventions following the trauma.
According to our results, no significant relationship was found between survivor's guilt, the time duration after the earthquake, the existence of psychiatric disorder history before the earthquake, and the place to stay in the first month after the earthquake. However, a weak but statistically significant negative correlation was found between age and survivor guilt, suggesting that younger individuals experienced higher levels of guilt. This finding is consistent with negative correlations between age and intrusion and avoidance and align with previous research showing that younger individuals often exhibit more intense trauma responses [54]. These age-related differences underscore the necessity of age-sensitive interventions in the aftermath of large-scale trauma. An important point we seek to emphasize is that post-traumatic changes in living conditions, rather than pre-existing factors like living conditions before the earthquake and conditions during the initial phase of the earthquake, are significantly associated with the severity of survivor guilt and PTSD symptoms. There was no relation between the time duration after the earthquake and PTSD symptom level and survivor guilt. This finding may be related to the timing of our assessment, which was conducted approximately nine months after the earthquake. Trauma responses should be examined at both earlier and later stages following the traumatic event to better understand the relationship between time duration and survivor guilt and PTSD.
- On page 13, the Limitations section. The authors should show official figures of disaster in the ten most affected cities of Turkey, to give us an idea of how representative the collected sample is.
We revised the Limitations section to clarify the issue of representativeness by explicitly stating that, although participants were recruited from the ten most severely affected provinces, the sample was not proportional to population size or the distribution of disaster impact.
A key limitation of the study concerns the sample size and its representativeness. Although participants were recruited from the ten provinces most severely affected by the earthquakes, the study population was not proportional to population size or the distribution of disaster impact across these regions. Additionally, it is essential to underscore that our investigation did not extend to an examination of the broader trauma histories of the participants and, notably, lacked a comparative analysis with a control group unexposed to the earthquake. The relatively modest sample size and insufficient qualified population distribution limit the findings' generalizability. Furthermore, the study focused entirely on individuals exposed to the disaster and did not include a control group or an assessment of participants’ history of another trauma. Future research should aim to use larger, more representative samples and include comparison groups, as addressing these methodological gaps may yield more substantial and generalizable insights.
Reviewer 3 Report
Comments and Suggestions for Authors
This article mainly explores the relationship between post-traumatic stress disorder (PTSD), survivor guilt, and pessimism schema among earthquake survivors, and explores the mediating role of survivor guilt in the relationship between early maladaptive schemas (EMS) and PTSD severity, providing important theoretical basis for understanding the pathogenesis of PTSD and developing effective treatment interventions. However, there remains room for enhancement in this article, and the following suggestions are respectfully offered for consideration:
- In the “Introduction”, the discussion on the relationship between EMS and PTSD is relatively broad, and there is insufficient review of previous research cases. Moreover, there is insufficient analysis on the mediating role of survivor guilt in EMS and PTSD.
- The Introduction section does not clearly point out the shortcomings in current research on the relationship between PTSD, EMS, and survivor guilt, and how this study will improve these shortcomings.
- The introduction lacks comprehensive and sufficient reasons for investigating the proposed research question.
- In “Materials and Methods”, the article mentions the use of BDI, BAI scales and tools, but does not fully explain the basis for selecting these specific evaluation indicators. For example, why choose these scales instead of other similar scales? How reliable and valid are they?
- In the “Results” section, it is recommended to add a visual explanation of the statistical data to facilitate understanding by non-specialist reads.
- Suggest considering adding robustness tests to confirm the stability of the main findings.
- Suggest adding heterogeneity analysis. For example, age, gender, severity of earthquakes, etc.
- The discussion section did not sufficiently engage in dialogue with existing research. Although the author reviewed some existing literature, they did not explore whether the conclusions found in this study have been found in other studies as well? Or are these relationships different in different contexts?
- Have the entire text reviewed for language quality to ensure consistent terminology and clear expression. Pay particular attention to avoiding overly complex sentences and maintaining clarity and conciseness. Also, ensure proper punctuation usage.
- Update the references to include the latest relevant literature, ensuring all cited materials are current and most representative.
Author Response
We would like to thank you for your valuable evaluation and comments. We have made the necessary changes emphasized by the reviewers. Reviewer comments (in italics), our response to reviewers, and the changes made in light of reviewer comments (in bold) are provided below.
This article mainly explores the relationship between post-traumatic stress disorder (PTSD), survivor guilt, and pessimism schema among earthquake survivors, and explores the mediating role of survivor guilt in the relationship between early maladaptive schemas (EMS) and PTSD severity, providing important theoretical basis for understanding the pathogenesis of PTSD and developing effective treatment interventions. However, there remains room for enhancement in this article, and the following suggestions are respectfully offered for consideration:
- In the “Introduction”, the discussion on the relationship between EMS and PTSD is relatively broad, and there is insufficient review of previous research cases. Moreover, there is insufficient analysis on the mediating role of survivor guilt in EMS and PTSD.
We added to the Introduction section the following part.
We expanded the Introduction section to include a more focused review of previous empirical studies that specifically examine the relationship between early maladaptive schemas (EMS) and PTSD, and the mediation role of EMS. These additions provide a stronger rationale for our model and address the identified gaps.
A recent review highlighted that nine Early Maladaptive Schemas related to the Disconnection/Rejection and Impaired Autonomy/Performance domains are commonly observed among trauma-experienced populations [7]. A study examining the relationship between childhood trauma and Complex PTSD reported a mediating role for EMSs. The study particularly focused on the total EMS score and the Disconnection and Impaired Autonomy schema domains [8].
To our knowledge, no study has yet examined the relationship between early maladaptive schemas, survivor guilt, and PTSD. Addressing this gap in the literature may contribute to the identification of appropriate therapeutic targets within schema therapy and other treatment modalities that focus on cognition, emotion, and experiential processing. As a key research question in our study, we investigated the mediating role of survivor guilt in the relationship between Early Maladaptive Schemas (EMS) and PTSD. Survivor guilt may serve as a cognitive-affective bridge between underlying EMS and PTSD symptoms. Individuals with activated and elevated early maladaptive schemas may be susceptible to experiencing guilt after surviving a disaster, which may, consequently, intensify trauma-related distress and symptom severity.
- The Introduction section does not clearly point out the shortcomings in current research on the relationship between PTSD, EMS, and survivor guilt, and how this study will improve these shortcomings.
- 3. The introduction lacks comprehensive and sufficient reasons for investigating the proposed research question.
The revisions and explanations provided in response to suggestion 1 also encompass suggestions 2 and 3
- In “Materials and Methods”, the article mentions the use of BDI, BAI scales and tools, but does not fully explain the basis for selecting these specific evaluation indicators. For example, why choose these scales instead of other similar scales? How reliable and valid are they?
We have explained the rationale for selecting the BDI and BAI in our manuscript. These self-report measures were chosen due to their widespread use, ease of administration, strong psychometric properties, and established validity and reliability in both international and Turkish populations. These characteristics made them appropriate tools for assessing depression and anxiety symptoms in our sample of earthquake survivors. The relevant explanation has been added to the Materials and Methods section
Beck Depression Inventory-II (BDI) was used to measure the severity of survivors' depression. It has 21 items, and each item is scored on a four-point scale ranging from 0 (absent) to 3 (severe) [27]. In the Turkish adaptation study, participants were asked to evaluate their depressive symptoms based on experiences during the past week. A total score of 17 or higher was accepted as the cut-off point for clinical depression [28]. The BDI has demonstrated high internal consistency and is widely used in clinical populations due to its robust psychometric properties. As a self-report measure, it enables practical and time-efficient application.
Beck Anxiety Inventory (BAI); was used to measure the severity of survivors' self-reported anxiety by administering it. It is a 4-point Likert-type questionnaire, with 21 items [29]. In the Turkish adaptation study, participants were instructed to assess their symptoms based on the past week, and a cut-off score of 16 was established [30]. The Turkish version of the BAI has been demonstrated to be a valid and reliable instrument across various populations, and it was considered both appropriate and easy to administer for assessing anxiety in the disaster-affected study sample.
- In the “Results” section, it is recommended to add a visual explanation of the statistical data to facilitate understanding by non-specialist reads.
Based on your suggestion, we highlighted significant data to improve visual clarity in the tables and separated continuous and categorical data. We would be happy to make any changes if you have any specific suggestions.
- Suggest considering adding robustness tests to confirm the stability of the main findings.
- Suggest adding heterogeneity analysis. For example, age, gender, severity of earthquakes, etc.
Answer to suggestions 6 and 7
We accept the value of these additional analyses. Still, the primary objective of our study was to explore the potential mediating role of survivor guilt between early maladaptive schemas and PTSD symptom severity. Given the present study's specific aims, sample size, and cross-sectional design, we maintained a focused and relatively limited analytical approach. Additional subgroup or sensitivity analyses may extend beyond the purpose of this manuscript. Nonetheless, we acknowledged this limitation and emphasized the need for such analyses in future research in the revised version.
Furthermore, due to the cross-sectional design and the current sample characteristics (like sample size), robustness or heterogeneity analyses (e.g., by age, gender, etc.) were not performed. In future studies conducted with larger samples and longitudinal designs, the use of such analyses would be beneficial for exploring subgroup differences and confirming the stability of the observed relationships.
- The discussion section did not sufficiently engage in dialogue with existing research. Although the author reviewed some existing literature, they did not explore whether the conclusions found in this study have been found in other studies as well? Or are these relationships different in different contexts?
We agree that a comprehensive engagement with existing literature strengthens the discussion. However, we would like to note that the specific relationship among early maladaptive schemas, survivor guilt, and PTSD remains a relatively underexplored area in the current literature. Despite this limitation, we have made every effort to contextualize our findings within the available research and provide a thoughtful discussion.
In line with your suggestion, we expanded the discussion section by incorporating newly identified relevant references and drawing clearer connections between our results and existing theoretical and empirical studies.
- Have the entire text reviewed for language quality to ensure consistent terminology and clear expression. Pay particular attention to avoiding overly complex sentences and maintaining clarity and conciseness. Also, ensure proper punctuation usage.
We thoroughly revised the manuscript to improve language quality, clarification, and to correct punctuation errors
- Update the references to include the latest relevant literature, ensuring all cited materials are current and most representative.
The discussion section was partially expanded. Additions were made from relevant literature.
Lian, A. E. Z.; Chooi, W. T.; Bono, S. A. (2023). A systematic review investigating the early maladaptive schemas (EMS) in Individuals with trauma experiences and PTSD. European Journal of Trauma & Dissociation, 7(1), 100315.
.Vasilopoulou, E.; Karatzias, T.; Hyland, P.; Wallace, H.; Guzman, A. (2019). The Mediating Role of Early Maladaptive Schemas in the Relationship between Childhood Traumatic Events and Complex Posttraumatic Stress Disorder Symptoms in Older Adults (>64 Years). Journal of Loss and Trauma, 25(2), 141–158. https://doi.org/10.1080/15325024.2019.1661598
Kip A,; Diele J; Holling H; Morina N. (2022) The relationship of trauma-related guilt with PTSD symptoms in adult trauma survivors: a meta-analysis. Psychol Med, 52(12):2201-2211. doi: 10.1017/S0033291722001866
Dell'Osso L, Carmassi C, Massimetti G, Stratta P, Riccardi I, Capanna C, Akiskal KK, Akiskal HS, Rossi A. (2013). Age, gender and epicenter proximity effects on post-traumatic stress symptoms in L'Aquila 2009 earthquake survivors. J Affect Disord, 5;146(2):174-80. doi: 10.1016/j.jad.2012.08.048.
Round 2
Reviewer 2 Report
Comments and Suggestions for Authors
Congratulations to the authors; they did a spectacular job of improving the manuscript.
Reviewer 3 Report
Comments and Suggestions for Authors
The revised manuscript has significantly improved through the incorporation of suggested changes and additional clarifications, and my concerns have been adequately addressed. I recommend this paper for publication in its current form.